# Factors associated with respectful maternity care during hospital deliveries: A cross-sectional study in Bangladesh

**Fahmida Islam Chowdhury[1]\*, Moshammat Zebunnessa[1], Mortuza Begum[1], Fahmida Shirin[1], Musarrat Naz[2], Samira Amir Chowdhury[1], Hasnatul Jannat[1], Selina Parvin[3], Md. Abdur Rafi[4], Mohammad Jahid Hasan[5]\***

1 Department of Obs & Gynae, Chittagong Medical College, Chittagong, Bangladesh, 2 Department of Obs & Gynae, Banshkhal Upazila Health Complex, Chittagong, Bangladesh, 3 Department of Obs & Gynae, Centre for Specialized Care & Research Hospital, Chittagong, Bangladesh, 4 Pi Research & Development Center, Dhaka, Bangladesh, 5 Tropical Disease and Health Research Center, Dhaka, Bangladesh

\* fahmidaislam31@yahoo.com (FIC); dr.jahid61@gmail.com, jahid.hasan@tdhrc.org (MJH)

## Abstract

### Introduction

Respectful provision of care is an integral component of quality maternity care service. The objective of the present study was to assess the status of respectful maternity care and its associated factors in public and private hospitals in Bangladesh.

### Methods

A cross-sectional study was conducted at a tertiary care public hospital and a tertiary care private hospital in Chittagong, Bangladesh from October 2023 to September 2024. Face-to-face interviews using a structured questionnaire was conducted to collect data from postnatal women. Respectful maternity care was measured using a validated 15-item tool with four domains (friendly, abuse-free, timely, and discrimination-free care). Logistic regression analysis was used to identify factors associated with respectful maternity care.

### Results

A number of 264 postnatal women from the public hospital and 334 from the private hospital were included in the study. Overall, 55.5% of them received respectful maternity care, with significant differences between public (33.7%) and private (72.8%) hospitals (p-value <0.001). Women in private hospitals reported higher standards across all domains, with the largest disparity in discrimination-free care (98% in private vs. 56% in public hospitals, p < 0.001). Logistic regression showed that women in private hospitals (adjusted odds ratio, aOR 18.10; 95% confidence interval, CI 8.43–42.0), those with facility-level referrals (aOR 2.88; 95% CI 1.59–5.31), and

**Data availability statement:** All relevant data are within the manuscript and its Supporting Information files.

**Funding:** The author(s) received no specific funding for this work.

**Competing interests:** The authors declare that they have no competing interests.

cesarean deliveries (aOR 2.45; 95% CI 1.26–5.07) were significantly more likely to receive respectful maternity care.

## Conclusion

Respectful maternity care was significantly more likely among women delivering in private hospitals, through facility-level referrals, and by cesarean section, indicating gaps in public hospital practices that require attention to ensure respectful care for all mothers.

---

## Introduction

Reducing maternal mortality has consistently been a global public health priority, particularly in low- and middle-income countries. The United Nations Sustainable Development Goals (SDGs) set a global target for the maternal mortality ratio (MMR) to reduce maternal deaths to fewer than 70 per 100,000 live births by 2030 [1].

In Bangladesh, a lower-middle-income country in South Asia, the MMR has decreased significantly over the past few decades, from 434 per 100,000 live births in 2000–173 per 100,000 live births in 2017 [2]. Despite these achievements, the MMR in this country remains above the SDG target, highlighting the necessity of ongoing efforts to improve maternal health outcomes. One critical factor in lowering the maternal mortality is increasing the proportion of facility-based deliveries, where skilled birth attendants (SBAs) are available to deliver timely and potentially life-saving care [3,4]. In Bangladesh, facility-based deliveries have seen a marked increase, rising from 9% in 2004 to 65% in 2022 [5]. However, simply increasing facility-based delivery rates is insufficient to ensure high-quality maternal care. Even when evidence-based clinical care is provided, it cannot be classified as quality care if it is not delivered in a respectful manner [6].

Effective maternal care must be delivered in a manner that is safe, effective, timely, efficient, equitable, and centered around the patients' needs [7,8]. The World Health Organization's (WHO) Framework for improving quality of care for women during childbirth highlighted the importance of women's experiences as a vital component of quality care, emphasizing that their experiences are just as critical as the clinical care provided [8]. Within this framework, respectful maternity care (RMC) is recognized as a critical component of high-quality maternal care. The concept of respectful maternity care refers to care that preserves the dignity, privacy, and confidentiality of the patient; ensures freedom from harm and mistreatment; and enables informed choice and continuous support throughout labor and childbirth [9]. When women experience disrespect and abuse in health facilities, this can discourage them from seeking future facility-based care, subsequently putting them at greater risk for complications that contribute to maternal mortality. However, in many low-resource settings, disrespect and abuse during childbirth are widespread and, to some extent, normalized [10,11].

In South Asian countries, women who access facility-based maternal services, frequently encounter high levels of disrespect, abuse, and mistreatment [12]. In Bangladesh, almost half of women are unable to maintain privacy during delivery, and based on their experiences in health facilities, they would become unwilling to return to the same facilities for future childbirth [13]. Such experiences indicate significant gaps in maternity care quality, as they deter women from seeking care in health facilities, which ultimately compromises efforts to reduce maternal mortality.

The healthcare system of Bangladesh has multiple stakeholders including public, private, and NGO-based sectors, all of which play crucial roles in providing maternal health services. Over recent years, there has been an increasing dependence on the private sector for childbirth services in Bangladesh, with more than half of institutional deliveries now occurring in private hospitals [14]. Considering this situation, it is essential to assess the quality of maternity care provided in both public and private hospitals to ensure that all women have access to high-quality care regardless of the facility type. Assessing the status of respectful maternity care could provide an important indicator of maternity care quality across these hospitals. Despite its significance, however, there is limited evidence regarding the status of respectful maternity care in public and private hospitals in Bangladesh.

Hence, the present study aimed to assess the status of respectful maternity care in public and private hospitals in Chittagong, Bangladesh.

## Methods

### Study design and setting

We conducted a facility-based cross-sectional study in a public and a private hospital in Chittagong (also locally known as Chattogram), Bangladesh from October 2023 to September 2024. As the country's second-largest division, Chittagong offers a diverse population with varied healthcare needs. We selected Chittagong Medical College Hospital (CMCH), a 2,200-bed public tertiary care hospital, and CSCR Hospital, a private tertiary care hospital as our study site.

### Study participants

We considered women admitted to either hospital for childbirth, including those undergoing normal vaginal delivery (NVD) and cesarean sections (CS) as our study population. The sample size was calculated using the following formula: $n = z^2 p(1-p)/d^2$, where: $z = 1.96$ for a 95% confidence level, $p = 0.5$, assuming that 50% of women received respectful maternity care, and $d = 0.05$, the margin of error. To account for potential clustering by hospital type, we applied a design effect of 1.5. We added 10% to account for possible non-responses, resulting in a final sample size of 640 participants.

We used a purposive sampling method to include study participants by reviewing the daily delivery records of the hospitals of previous 24 hours. We excluded patients who were severely ill or unable to respond. If a selected patient was unavailable, we approached the next eligible patient.

### Data collection

We conducted face-to-face interviews with the participants within 24 hours of their delivery using a structured questionnaire. The questionnaire was divided into two sections:

(i)  **Socio-Demographic and Obstetric Information:** This section included participants' demographic details, such as age, education, residence (urban/rural), and occupation, as well as obstetric details, such as parity (number of previous births), number of antenatal care (ANC) visits, referral status, length of labor, mode of delivery (vaginal or cesarean), presence of a companion during labor (e.g., husband or relative), and the gender of the labor attendant (doctor, nurse, or midwife).

(ii) **Respectful Maternity Care Information:** This section assessed the level of respectful maternity care using a validated 15-item tool developed by Sheferaw et al. [15]. This tool included four domains: friendly care (7 items)

measuring respectful communication, abuse-free care (3 items) measuring the verbal or physical abuse, timely care (2 items) evaluating the promptness of care, and discrimination-free care (3 items) assessing freedom from discrimination based on socioeconomic or demographic factors.

The tool used a five-point Likert scale, from 'strongly disagree' (1) to 'strongly agree' (5). We categorized responses of 'strongly agree' and 'agree' as 'Yes' (indicating respectful care), and responses of 'strongly disagree', 'disagree', and 'neutral' as 'No' (indicating disrespect or abuse) for positively worded items. Six statements were negatively worded, and their responses were reverse coded before analysis to ensure consistency in interpretation.

The sociodemographic part of the questionnaire was prepared by our study team and the respectful maternity care part was adopted from Sheferaw et al. [15]. The questionnaire was prepared in English and then translated into Bangla using back-translation method by two independent translators. We did not do formal validation of the Bangla translation of the questionnaire. However, it was pretested on 30 women who had recently delivered at our selected hospitals, and required adjustments were made based on their feedback to ensure cultural and linguistic appropriateness. In our study, the tool demonstrated good internal consistency, with a Cronbach's alpha of 0.818. The questionnaire is provided in the 'S1 File'.

The study protocol was reviewed and approved by Public Health Foundation, Bangladesh (REF no: PHFBD-ERC-SG15/2023). Informed written consent was obtained from the patient or legal guardian.

## Statistical analysis

All the statistical analyses were done using R version 4.4.1. We used descriptive statistics, such as frequencies and percentages for categorical variables and means and standard deviations for continuous variables, to summarize the participants' demographic and clinical characteristics.

Level of respectful maternity care was assessed separately for both public and private hospitals. We classified women as having received respectful maternity care if all items were coded as Yes; if any item was coded as No, the woman was considered to have not received respectful maternity care. For the six negatively worded items, responses were reverse coded prior to dichotomization so that "disagree" or "strongly disagree" indicated respectful care. This ensured that across all items, a response coded as "Yes" consistently represented respectful care, and "No" indicated disrespect or abuse. This categorization was applied for the overall scale as well as for each of the specific domains.

To examine the factors associated with receiving respectful maternity care, we conducted multivariable logistic regression analysis. Our outcome variable was the binary indicator for respectful maternity care (received vs. not received). We included socio-demographic and obstetric factors as independent variables: age group, educational attainment, residence, occupation, parity, number of ANC visits, hospital type (public or private), referral status, length of labor, mode of delivery, presence of a companion during labor, and gender of the labor attendant. We calculated adjusted odds ratios (aOR) with 95% confidence intervals (CI) to quantify the association between each predictor and the likelihood of receiving respectful care. We considered associations statistically significant if the p-value was below 0.05.

## Results

### Participants' characteristics

A total of 598 women participated in the study, with 264 from the public hospital and 334 from the private hospital. The mean age of the participants was 26.7 years (SD 5.2 years). While approximately two-thirds of the women in the public hospital came from rural areas, this proportion was significantly lower at the private hospital (27%).

Nearly two-thirds of the participants were multiparous, and a majority had received at least four ANC visits. However, there was a significant difference in the uptake of ANC visits between the hospitals– 92% of women in the private hospital reported at least four ANC visits, compared to only 49% in the public hospital. The referral patterns also differed between the two facilities; nearly half of the women in the public hospital (46%) were referred by another facility, whereas most

women in the private hospital (97.6%) were self-referred or had planned their delivery at the hospital. Delivery mode also varied between the hospitals. The rate of cesarean section was higher in the public hospital, where 81% of women underwent cesarean delivery, compared to only 31% in the private hospital. The presence of a companion, such as a husband or relative, during delivery was more common in the private hospital (93%) than in the public hospital (58%) (**Table 1**).

### Status of respectful maternity care

Overall, 55.5% of women received respectful maternity care, with a significant difference between government and private hospitals (33.7% and 72.8%, respectively, p-value <0.001) (**Table 2**). For the domain of friendly care, 85.5% of women reported receiving such care, with 78.0% in government hospitals and 91.3% in private hospitals. In the domain of abuse-free care, 80.1% of women reported no abuse during maternity care, with a distribution of 69.7% in government hospitals and 88.3% in private hospitals. For timely care, 77.9% of women received care in a timely manner, with 72.0% in government hospitals and 82.6% in private hospitals. In the domain of discrimination-free care, 79.4% of women experienced no discrimination, with significant differences between hospital types: 56.1% in government hospitals and 97.9% in private hospitals, p-value <0.001 (**Fig 1**).

### Factors associated with respectful maternity care

In the multiple logistic regression analysis, we found that women who delivered in the private hospital had higher odds of receiving respectful maternity care compared to those who delivered in the public hospital (adjusted odds ratio, aOR 18.10, 95% confidence interval, CI 8.43, 42.0). Besides, women with a facility-level referral had nearly threefold higher odds of receiving respectful care compared to those with self-referral (aOR 2.88; 95% CI 1.59, 5.31). Additionally, women who underwent cesarean section were more likely to receive respectful care than those who had a normal vaginal delivery (NVD) (aOR 2.45; 95% CI 1.26, 5.07) (**Table 3**).

## Discussion

Our study found that approximately one-third of the women delivering in public hospitals and two-thirds of the women delivering in private hospitals in Chittagong, Bangladesh, received respectful maternity care, while the remainder experienced at least one form of disrespect or abuse. In terms of specific care domains, around 85% reported receiving friendly care, 78% received timely care, and 79% received discrimination-free care. Additionally, about 80% of women received abuse-free care, meaning that roughly one in five women experienced some form of verbal or physical abuse from healthcare providers during delivery. A significant difference was observed between public and private hospitals, with women in private hospitals receiving more respectful care across all domains. The most substantial disparity was observed in the domain of discrimination-free care, where approximately half of the women in public hospitals experienced discrimination, while most women in private hospitals reported receiving discrimination-free care.

Evidence regarding respectful maternity care in Bangladesh is limited. However, some empirical data suggest that respectful maternity care remains a neglected issue within the country's healthcare context. Although approximately 73% of maternity care facilities in Bangladesh report having policies and processes in place to identify client abuse, the practical implementation of these policies is limited. A majority of healthcare providers are not adequately trained in respectful maternity care, and structural challenges prevent effective care delivery. According to one study, nearly half of the women in Bangladesh were unable to maintain their privacy during labor, and around 7% reported experiencing verbal abuse during delivery. Based on their experiences in healthcare facilities, half of the women stated that they would not return to the same facility for future childbirth [13]. Another study in Bangladesh reported that more than half of mothers experienced disrespect and abuse during delivery [16]. However, from the perspective of service providers, respectful care may be underestimated as an issue, as most healthcare providers (including nurses, midwives, and doctors) reported that they believed they provided respectful care [13]. The lack of adequate training in respectful maternity care appears to be a

**Table 1. Sociodemographic and obstetric characteristics of the participants (n = 598).**

| Characteristics | Overall, n = 598 | Public hospital, n = 264 | Private hospital, n = 334 |
|---|---|---|---|
| Age (years), mean (SD) | 26.70 (5.21) | 26.00 (5.63) | 27.23 (4.81) |
| Age group (years) | | | |
| 18-24 | 208 (35.62) | 106 (41.90) | 102 (30.82) |
| 25-34 | 330 (56.51) | 120 (47.43) | 210 (63.44) |
| ≥35 | 46 (7.88) | 27 (10.67) | 19 (5.74) |
| Educational qualification | | | |
| No formal education | 21 (3.52) | 17 (6.49) | 4 (1.20) |
| Primary | 17 (2.85) | 10 (3.82) | 7 (2.10) |
| Secondary | 130 (21.81) | 82 (31.30) | 48 (14.37) |
| Higher secondary | 272 (45.64) | 103 (39.31) | 169 (50.60) |
| University graduate | 156 (26.17) | 50 (19.08) | 106 (31.74) |
| Residence | | | |
| Urban | 326 (54.88) | 85 (32.44) | 241 (72.59) |
| Rural | 268 (45.12) | 177 (67.56) | 91 (27.41) |
| Occupation | | | |
| Homemaker | 540 (92.15) | 244 (93.85) | 296 (90.80) |
| Job holder | 33 (5.63) | 15 (5.77) | 18 (5.52) |
| Business | 13 (2.22) | 1 (0.38) | 12 (3.68) |
| Parity | | | |
| Primipara | 220 (36.97) | 96 (36.64) | 124 (37.24) |
| Multipara | 375 (63.03) | 166 (63.36) | 209 (62.76) |
| Number of ANC visit | | | |
| <4 | 159 (26.90) | 133 (51.35) | 26 (7.83) |
| ≥4 | 432 (73.10) | 126 (48.65) | 306 (92.17) |
| Referral status | | | |
| Previously planned/self-referral | 465 (78.28) | 142 (53.99) | 323 (97.58) |
| Facility referral | 129 (21.72) | 121 (46.01) | 8 (2.42) |
| Length of labor (hours) | | | |
| <12 | 293 (56.67) | 198 (79.52) | 95 (35.45) |
| ≥12 | 224 (43.33) | 51 (20.48) | 173 (64.55) |
| Missing data | 81 | 15 | 66 |
| Mode of delivery | | | |
| Normal vaginal delivery | 274 (45.90) | 48 (18.18) | 226 (67.87) |
| Cesarean section | 321 (53.77) | 215 (81.44) | 106 (31.83) |
| Companion (husband/relative) during labor | | | |
| Yes | 420 (76.78) | 146 (57.94) | 274 (92.88) |
| No | 127 (23.22) | 106 (42.06) | 21 (7.12) |
| Missing data | 51 | 12 | 39 |
| Gender of labor attendant | | | |
| Male | 15 (2.56) | 13 (5.14) | 2 (0.60) |
| Female | 571 (97.44) | 240 (94.86) | 331 (99.40) |

**Table 2. Characteristics of the participants according to respectful maternity care.**

| Characteristics | Respectful maternity care | | p-value |
|---|---|---|---|
| | Yes, n = 332 | No, n = 266 | |
| Age group (years) | | | 0.062 |
| 18-24 | 104 (50.00) | 104 (50.00) | |
| 25-34 | 199 (60.30) | 131 (39.70) | |
| ≥35 | 25 (54.35) | 21 (45.65) | |
| Hospital type | | | <0.001 |
| Public | 89 (33.71) | 175 (66.29) | |
| Private | 243 (72.75) | 91 (27.25) | |
| Educational qualification | | | 0.157 |
| No formal education | 12 (57.14) | 9 (42.86) | |
| Primary | 9 (52.94) | 8 (47.06) | |
| Secondary | 60 (46.15) | 70 (53.85) | |
| Higher secondary | 157 (57.72) | 115 (42.28) | |
| University graduate | 94 (60.26) | 62 (39.74) | |
| Residence | | | 0.005 |
| Urban | 198 (60.74) | 128 (39.26) | |
| Rural | 132 (49.25) | 136 (50.75) | |
| Occupation | | | 0.262 |
| Homemaker | 294 (54.44) | 246 (45.56) | |
| Job holder | 19 (57.58) | 14 (42.42) | |
| Business | 10 (76.92) | 3 (23.08) | |
| Parity | | | 0.730 |
| Primipara | 120 (54.55) | 100 (45.45) | |
| Multipara | 210 (56.00) | 165 (44.00) | |
| Number of ANC visit | | | 0.195 |
| <4 | 61 (38.36) | 98 (61.64) | |
| ≥4 | 267 (61.81) | 165 (38.19) | |
| Referral status | | | 0.004 |
| Previously planned/self-referral | 272 (58.49) | 193 (41.51) | |
| Facility referral | 57 (44.19) | 72 (55.81) | |
| Length of labor (hours) | | | 0.004 |
| <12 | 143 (48.81) | 150 (51.19) | |
| ≥12 | 138 (61.61) | 86 (38.39) | |
| Mode of delivery | | | 0.009 |
| Normal vaginal delivery | 167 (60.95) | 107 (39.05) | |
| Cesarean section | 164 (51.09) | 157 (48.91) | |
| Companion (husband/relative) during labor | | | 0.002 |
| Yes | 241 (57.38) | 179 (42.62) | |
| No | 53 (41.73) | 74 (58.27) | |
| Gender of labor attendant | | | 0.462 |
| Male | 7 (46.67) | 8 (53.33) | |
| Female | 321 (56.22) | 250 (43.78) | |

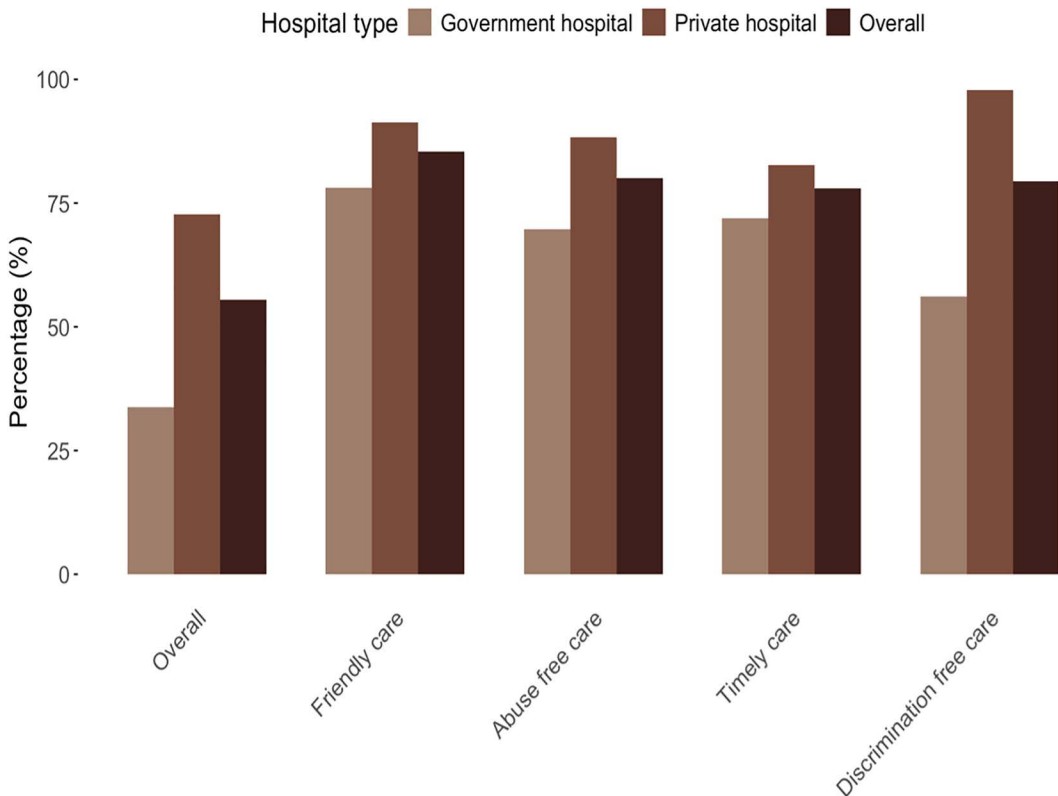

**Fig 1. Status of respectful maternity care among participants (n = 598).**

significant barrier. Establishing care standards, improving training, and creating an enabling environment for quality maternal care are necessary to increase the implementation of respectful maternity care practices [17].

In this study, women in private hospitals reported higher rates of respectful maternity care compared to those in public hospitals. Notably, around half of the women delivering in public hospitals reported experiencing discrimination based on their socioeconomic or other contextual factors, whereas in private hospitals, only 2% of women reported discrimination. This finding is consistent with a recent study from Bangladesh, which observed that disrespect and abuse were more prevalent in public healthcare facilities (73%) compared to private facilities (40%) [16]. It is important to consider that public hospitals in Bangladesh typically serve a diverse socioeconomic population, largely consisting of poorer individuals, while private hospitals generally cater to a more homogeneous, affluent clientele. However, as this study was conducted in a single public and private hospital, the findings may not be representative of the entire country. Nevertheless, the overall responsiveness of public facilities in Bangladesh is generally low. Many care recipients have rated the responsiveness of these facilities poorly, particularly with regard to prompt attention, dignity, clear communication, and confidentiality [18]. Furthermore, the concept of respectful maternity care is rooted in basic human rights, which should not be compromised.

Another key finding from our study was that women with a facility-level referral reported better respectful maternity care compared to those who were self-referred. In Bangladesh, the healthcare referral system is not well established, and a large portion of patients are self-referred to tertiary care facilities [19], which was also observed in this study. When patients self-refer to tertiary care hospitals, there is often a lack of proper medical history and guidance for healthcare providers, which can hinder quality care, especially given the high patient burden in these settings.

**Table 3. Factors associated with respectful maternity care among participants (logistic regression models).**

| Characteristics | cOR (95% CI) | aOR (95% CI) |
|---|---|---|
| Age group (years) | | |
| 18-24 | 1 | 1 |
| 25-34 | 1.52 (1.07, 2.16)* | 1.27 (0.79, 2.03) |
| ≥35 | 1.19 (0.63, 2.28) | 1.19 (0.50, 2.85) |
| Hospital type | | |
| Public | 1 | 1 |
| Private | 5.25 (3.71, 7.49)* | 18.1 (8.43, 42.0)* |
| Educational qualification | | |
| No formal education | 1 | 1 |
| Primary | 0.84 (0.23, 3.08) | 0.30 (0.06, 1.52) |
| Secondary | 0.64 (0.25, 1.62) | 0.44 (0.13, 1.43) |
| Higher secondary | 1.02 (0.41, 2.50) | 0.45 (0.14, 1.42) |
| University graduate | 1.14 (0.44, 2.85) | 0.45 (0.13, 1.51) |
| Residence | | |
| Urban | 1 | 1 |
| Rural | 0.63 (0.45, 0.87) | 1.38 (0.85, 2.26) |
| Occupation | | |
| Homemaker | 1 | 1 |
| Job holder | 1.14 (0.56, 2.35) | 1.13 (0.45, 2.90) |
| Business | 2.79 (0.84, 12.5) | 1.30 (0.34, 6.34) |
| Parity | | |
| Primipara | 1 | 1 |
| Multipara | 1.06 (0.76, 1.48) | 1.30 (0.79, 2.12) |
| Number of ANC visit | | |
| <4 | 1 | 1 |
| ≥4 | 2.60 (1.79, 3.79)* | 1.52 (0.89, 2.60) |
| Referral status | | |
| Previously planned/self-referral | 1 | 1 |
| Facility referral | 0.56 (0.38, 0.83)* | 2.88 (1.59, 5.31)* |
| Length of labor (hours) | | |
| <12 | 1 | 1 |
| ≥12 | 1.68 (1.18, 2.40)* | 1.03 (0.59, 1.77) |
| Mode of delivery | | |
| Normal vaginal delivery | 1 | 1 |
| Cesarean section | 0.67 (0.48, 0.93)* | 2.45 (1.26, 5.07)* |
| Companion (husband/relative) during labor | | |
| Yes | 1 | 1 |
| No | 0.53 (0.35, 0.79)* | 1.65 (0.90, 3.09) |
| Gender of labor attendant | | |
| Male | 1 | 1 |
| Female | 1.47 (0.52, 4.24) | 0.53 (0.13, 2.01) |

cOR: crude odds ratio, aOR: adjusted odds ratio

*p-value <0.05

The mode of delivery also found as a significant predictor for respectful maternity care, with women who had cesarean sections reporting higher levels of respectful care compared to those who had normal vaginal deliveries. During cesarean sections, patients are generally more vulnerable, which may prompt healthcare providers to offer care with greater compassion and empathy. Another factor is that cesarean sections are typically conducted by doctors, while a majority of uncomplicated vaginal deliveries are performed by other medical staff, such as nurses or midwives, who may lack adequate training in respectful care in the context of Bangladesh. Previous studies have shown that the quality of care can be improved through the presence of facility mentors, who create enabling environments that promote higher standards of care [20].

Our findings signifies the need to strengthen respectful maternity care, especially in public hospitals, where poorer women are disproportionately affected by discrimination and abuse. Targeted training for nurses, midwives, and junior staff, along with accountability and monitoring systems, are essential. Strengthening referral pathways could improve continuity and responsiveness of care. Embedding respectful maternity care into national policies, quality assurance frameworks, and training curricula will help ensure equity, uphold women's rights, and improve maternal health outcomes in Bangladesh.

Our study had several limitations. First, being conducted in one government and one private hospital in Chittagong, the findings may lack generalizability to other settings of Bangladesh. Second, collecting data within the hospital environment could introduce social desirability bias, with participants possibly feeling pressured to provide positive responses or reluctant to report disrespectful or abusive care experiences. Although interviews were conducted in a private setting to reduce this bias, it may still be present. Third, data collection in the immediate postnatal period may have influenced responses, as some women might have been fatigued and unable to answer fully. Additionally, the scale used to measure respectful maternity care, while structured, may have limitations in capturing the full complexity of women's experiences. It may miss some behaviors, subtle forms of discrimination, and culturally specific aspects of respectful care, possibly underestimating care deficiencies. Fourth, by converting the original five-point Likert responses into a binary (Yes/No) variable, some nuances and gradations of women's experiences may have been lost, potentially leading to information loss. Finally, the study did not examine socioeconomic inequalities in respectful maternity care in detail, which may be critical for understanding disparities in care quality across different demographic groups. Our participants included a higher proportion of women with upper secondary and university education compared with the general female population of Bangladesh, which may limit the generalizability of our findings to less-educated groups. Besides, some significant shifts between crude and adjusted odds ratios, such as the reversal in association with facility referrals, may be influenced by the relatively small sample size and confounding factors, which should be considered when interpreting these results.

## Conclusions

In conclusion, our study highlighted a significant difference in respectful maternity care between public and private hospitals in Chittagong, Bangladesh. While a considerable proportion of women reported receiving friendly, timely, abuse-free, and discrimination-free care, those delivering in private hospitals consistently experienced higher standards of respectful maternity care across all assessed domains compared to their counterparts in public hospitals. Additionally, women who were institutionally referred and those who underwent cesarean sections were more likely to receive respectful care.

## Supporting information

**S1 File. Questionnaire.**
(PDF)

**S2 File. Data.**
(XLSX)

## Acknowledgments

The authors would like to express their sincere gratitude to Pi Research & Development Center, Dhaka-1100, Bangladesh (www.pirdc.org), for their help in manuscript revision and editing.

## Author contributions

**Conceptualization:** Fahmida Islam Chowdhury, Moshammat Zebunnessa, Mortuza Begum, Fahmida Shirin, Musarrat Naz, Samira Amir Chowdhury, Hasnatul Jannat, Selina Parvin, Md. Abdur Rafi, Mohammad Jahid Hasan.

**Data curation:** Musarrat Naz, Samira Amir Chowdhury, Hasnatul Jannat, Selina Parvin.

**Formal analysis:** Fahmida Shirin, Musarrat Naz, Hasnatul Jannat, Md. Abdur Rafi, Mohammad Jahid Hasan.

**Investigation:** Fahmida Islam Chowdhury, Moshammat Zebunnessa, Mortuza Begum, Samira Amir Chowdhury, Selina Parvin.

**Methodology:** Fahmida Islam Chowdhury, Moshammat Zebunnessa, Mortuza Begum, Hasnatul Jannat, Mohammad Jahid Hasan.

**Project administration:** Hasnatul Jannat.

**Resources:** Mortuza Begum, Mohammad Jahid Hasan.

**Software:** Musarrat Naz, Selina Parvin, Md. Abdur Rafi.

**Supervision:** Fahmida Islam Chowdhury, Moshammat Zebunnessa, Fahmida Shirin, Musarrat Naz, Md. Abdur Rafi, Mohammad Jahid Hasan.

**Writing – original draft:** Fahmida Islam Chowdhury, Moshammat Zebunnessa, Mortuza Begum, Fahmida Shirin, Musarrat Naz, Samira Amir Chowdhury, Hasnatul Jannat, Selina Parvin, Md. Abdur Rafi, Mohammad Jahid Hasan.

**Writing – review & editing:** Fahmida Islam Chowdhury, Moshammat Zebunnessa, Mortuza Begum, Fahmida Shirin, Musarrat Naz, Samira Amir Chowdhury, Selina Parvin, Md. Abdur Rafi, Mohammad Jahid Hasan.

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
