## [Decision Letter · Decision Letter 0]

12 Aug 2025

PONE-D-25-20339Respectful maternity care during hospital deliveries in Bangladesh: a comparative study between public and private hospitalPLOS ONE

Dear Dr. Hasan,

Thank you for submitting your manuscript to PLOS ONE. After careful consideration, we feel that it has merit but does not fully meet PLOS ONE’s publication criteria as it currently stands. Therefore, we invite you to submit a revised version of the manuscript that addresses the points raised during the review process.

**Before resubmitting a revised version, kindly ensure:****1. ** The manuscript has been reviewed by an English Language expert for improving the grammar quality.==============================

We look forward to receiving your revised manuscript.

Kind regards,

Paridhi Jha, PhD

Academic Editor

PLOS ONE

2. In the online submission form, you indicated that [Patient-level data will be available on request from the corresponding author.].

The following resources for replacing copyrighted map figures may be helpful

Additional Editor Comments (if provided):

Reviewers' comments:

Reviewer's Responses to Questions

**Comments to the Author**

1. Is the manuscript technically sound, and do the data support the conclusions?

Reviewer #1: Partly

Reviewer #2: Partly

Reviewer #3: Yes

2. Has the statistical analysis been performed appropriately and rigorously? 

Reviewer #1: Yes

Reviewer #2: I Don't Know

Reviewer #3: Yes

3. Have the authors made all data underlying the findings in their manuscript fully available?

Reviewer #1: Yes

Reviewer #2: No

Reviewer #3: Yes

4. Is the manuscript presented in an intelligible fashion and written in standard English?

Reviewer #1: Yes

Reviewer #2: Yes

Reviewer #3: Yes

5. Review Comments to the Author

Reviewer #1: The Title relate to a Comparative study - whilst the Manuscript refer to a Cross- Sectional study (In essence, a comparative study is more about drawing distinctions between groups, while a cross-sectional study is more about describing a situation across a population at a moment in time. That said, a cross-sectional study can also be comparative if it’s comparing subgroups within the population)

The reader hopes to find in the conclusion the goal that the Cross-sectional hoped to uncover such as in this research undertaking (The Objective of the study is to investigate the status of respectful maternity care and its associated factors in public and private hospitals in Bangladesh)

Notably, the Conclusion in the Abstract only refer to:

A study finding showing a significant difference in respectful maternity care between

public and private hospitals, with private facilities showing markedly higher care standards across domains. (This is generally a known fact).

In the Manuscript the Discussion 0n Page 10 Line 229 - 232 assert:

Discussion:

230 Our study found that approximately half of the women delivering in either public or private

231 hospitals in Chattagram, Bangladesh, received respectful maternity care, while the remainder

232 experienced at least one form of disrespect or abuse - (this is not consistent with the Conclusion - Suggest revisiting the Conclusion as the two groups compares a particular variable or outcome...RMC.

Importantly, the structured questionnaire ... Focuses on Status of respectful maternity care (RMC) - (hence the comparison between the two groups on these aspects) Figure 2: Status of respectful maternity care among participants (n = 598) depicts the status of RMC and its associated factors - hence, suggesting the Conclusion should congruently address such aspects.

Reviewer #2: The word “postpartum” is misspelled in multiple places (as “postpurtam” and similar spellings). Fix spelling throughout. (It is also acceptable to use “postnatal,” if preferred.)

I recognize that the city of Chittagong is colloquially known as Chottogram in Bangladesh, so the use of this name in the paper makes sense. However, most people in the world do not know that this city has two names, so at some point early in the paper, please explain that the two names refer to one city.

You also have Chottogram spelled in a variety of ways (Chottogram, Chattagram, Chattogram, etc.). I recognize these differences are basically moot when using Bangla script, but to avoid confusion among English-speakers, using a single spelling is essential (or replacing this everywhere with Chittagong). I have no opinion on which is best, but make sure one spelling is consistently applied throughout the document. (One exception: in all institutional/hospital/university names that include either Chittagong or Chottogram, please use the exact formal name, with whichever name and spelling the institution uses.)

The educational qualifications in Tables 1 and 2 do not align with the educational item on the survey. Please report these numbers using the same categories you used to collect the data.

In lines 180-182, you say “We classified women as having received respectful maternity care if all items were coded as ‘Yes’. If any item was coded as ‘No’, the woman was considered to have not received respectful maternity care.”

However, 6 items on the survey would be answered “yes” if DISRESPECTFUL maternity care occurred:

• Some healthcare providers slapped me during delivery for different reasons

• Some health workers shouted at me because I haven’t done what I was told to do

• I was kept waiting for a long time before receiving services

• Service provision was delayed due to the health facilities’ internal problem

• Some of the health workers do not treat me well because of some personal attribute

• Some health workers insulted me and my companions due to my personal attributes

Scoring for those items would have to count “no” as the presence of respectful maternity care. This is in contrast to all the other items in the survey, where answering “yes” would indicate respectful care. Scoring and statistics must account for the six reverse-scored items, or the results of your calculations would be incorrect.

It’s not clear how extensive this problem is. It's possible that you simply need to state more clearly how you did the scoring of responses before statistics were calculated. It’s also possible that essentially every finding on respectful maternity care in this paper is incorrect because of this scoring error.

In either case, it is likely fixable. Please clarify the issue, check all underlying raw data and every statistic throughout the paper to ensure that all are correct, and fix any miscalculations.

Reviewer #3: Thank you for submitting your manuscript. The following are some minor comments to help you improve the paper.

General & Title

Title: I suggest a revised title to better reflect the study's design and focus: "Factors associated with respectful maternity care during hospital deliveries: a cross-sectional study in Bangladesh."

Results Section

Clarity of Acronyms: For the first time they appear in both the abstract and the results section, please spell out aOR (Adjusted Odds Ratio) and 95% CI (95% Confidence Interval) before using the acronyms.

Figure 1: I recommend removing Figure 1 as it does not appear to be essential for the main findings of the paper.

Table 2: I suggest reorganizing your presentation of the data. Please remove the first two columns (Respectful maternity care) from Table 2. Instead, create a new table that presents all the predictor variables against the two outcome categories (respected vs. not respected), including p-values. This new table should be placed just before your logistic regression analysis.

Discussion Section

Limitations: Please include this point as one of key limitations of the study: the potential information loss that occurred when converting the original 5-point Likert scale into a binary (Yes/No) categorical variable.

Policy Implications: I recommend adding a section that discusses the practical policy implications of your findings, outlining how this research can inform future interventions or policies in Bangladesh.

6. PLOS authors have the option to publish the peer review history of their article (what does this mean? ). If published, this will include your full peer review and any attached files.

**Do you want your identity to be public for this peer review?** For information about this choice, including consent withdrawal, please see our Privacy Policy .

Reviewer #1: No

Reviewer #2: No

Reviewer #3: No

---

## [Author Response · Author response to Decision Letter 1]

5 Sep 2025

PONE-D-25-20339

Respectful maternity care during hospital deliveries in Bangladesh: a comparative study between public and private hospital

Reviewer #1:

The Title relate to a Comparative study - whilst the Manuscript refer to a Cross- Sectional study (In essence, a comparative study is more about drawing distinctions between groups, while a cross-sectional study is more about describing a situation across a population at a moment in time. That said, a cross-sectional study can also be comparative if it’s comparing subgroups within the population)

The reader hopes to find in the conclusion the goal that the Cross-sectional hoped to uncover such as in this research undertaking (The Objective of the study is to investigate the status of respectful maternity care and its associated factors in public and private hospitals in Bangladesh)

Notably, the Conclusion in the Abstract only refer to:

A study finding showing a significant difference in respectful maternity care between

public and private hospitals, with private facilities showing markedly higher care standards across domains. (This is generally a known fact).

In the Manuscript the Discussion 0n Page 10 Line 229 - 232 assert:

Discussion:

230 Our study found that approximately half of the women delivering in either public or private

231 hospitals in Chattagram, Bangladesh, received respectful maternity care, while the remainder

232 experienced at least one form of disrespect or abuse - (this is not consistent with the Conclusion - Suggest revisiting the Conclusion as the two groups compares a particular variable or outcome...RMC.

Importantly, the structured questionnaire ... Focuses on Status of respectful maternity care (RMC) - (hence the comparison between the two groups on these aspects) Figure 2: Status of respectful maternity care among participants (n = 598) depicts the status of RMC and its associated factors - hence, suggesting the Conclusion should congruently address such aspects.

Authors’ response:

We sincerely thank the reviewer for the thoughtful and constructive comments. We appreciate the careful reading and the emphasis on ensuring that the manuscript accurately reflects both the study design and its objectives.

First, regarding the study title, we acknowledge the reviewer’s observation that the original title suggested a comparative study, whereas the manuscript is based on a cross-sectional design. To address this, we have revised the title to clearly indicate that this is a cross-sectional study. The present version of the title is “Factors associated with respectful maternity care during hospital deliveries: a cross-sectional study in Bangladesh”

Second, we recognize the reviewer’s concern that the abstract conclusion previously highlighted only the differences in respectful maternity care (RMC) between public and private hospitals. While this comparison is important, it did not fully convey the study’s objective of assessing the overall status of RMC and its associated factors. In response, we have updated the abstract conclusion as:

“Respectful maternity care was significantly more likely among women delivering in private hospitals, through facility-level referrals, and by cesarean section, indicating gaps in public hospital practices that require attention to ensure respectful care for all mothers.”

Third, we addressed the inconsistency between the discussion and the abstract conclusion. The discussion had noted that

“Our study found that approximately one-third of the women delivering in public hospitals and two-thirds of the women delivering in private hospitals in Chittagong, Bangladesh, received respectful maternity care, while the remainder experienced at least one form of disrespect or abuse.”

Finally, we have ensured that the conclusion reflects the primary focus of our study, which is the status of RMC and its associated factors. These revisions also align the narrative with Figure 2 and the statistical analyses, providing a clear and accurate reflection of our study findings.

We sincerely thank the reviewer again for these valuable comments, which have strengthened the clarity, consistency, and scientific rigor of our manuscript.

Reviewer #2:

The word “postpartum” is misspelled in multiple places (as “postpurtam” and similar spellings). Fix spelling throughout. (It is also acceptable to use “postnatal,” if preferred.)

I recognize that the city of Chittagong is colloquially known as Chottogram in Bangladesh, so the use of this name in the paper makes sense. However, most people in the world do not know that this city has two names, so at some point early in the paper, please explain that the two names refer to one city.

You also have Chottogram spelled in a variety of ways (Chottogram, Chattagram, Chattogram, etc.). I recognize these differences are basically moot when using Bangla script, but to avoid confusion among English-speakers, using a single spelling is essential (or replacing this everywhere with Chittagong). I have no opinion on which is best, but make sure one spelling is consistently applied throughout the document. (One exception: in all institutional/hospital/university names that include either Chittagong or Chottogram, please use the exact formal name, with whichever name and spelling the institution uses.)

The educational qualifications in Tables 1 and 2 do not align with the educational item on the survey. Please report these numbers using the same categories you used to collect the data.

In lines 180-182, you say “We classified women as having received respectful maternity care if all items were coded as ‘Yes’. If any item was coded as ‘No’, the woman was considered to have not received respectful maternity care.”

However, 6 items on the survey would be answered “yes” if DISRESPECTFUL maternity care occurred:

• Some healthcare providers slapped me during delivery for different reasons

• Some health workers shouted at me because I haven’t done what I was told to do

• I was kept waiting for a long time before receiving services

• Service provision was delayed due to the health facilities’ internal problem

• Some of the health workers do not treat me well because of some personal attribute

• Some health workers insulted me and my companions due to my personal attributes

Scoring for those items would have to count “no” as the presence of respectful maternity care. This is in contrast to all the other items in the survey, where answering “yes” would indicate respectful care. Scoring and statistics must account for the six reverse-scored items, or the results of your calculations would be incorrect.

It’s not clear how extensive this problem is. It's possible that you simply need to state more clearly how you did the scoring of responses before statistics were calculated. It’s also possible that essentially every finding on respectful maternity care in this paper is incorrect because of this scoring error.

In either case, it is likely fixable. Please clarify the issue, check all underlying raw data and every statistic throughout the paper to ensure that all are correct, and fix any miscalculations.

Authors’ response:

We sincerely thank the reviewer for their careful reading and constructive comments, which have greatly helped improve the clarity and accuracy of our manuscript.

First, regarding the spelling of “postpartum/postnatal,” we have corrected all misspellings of “postpartum” (e.g., “postpurtam”) throughout the manuscript. Following the reviewer’s suggestion, we have consistently used “postnatal” to describe the period following delivery.

Second, concerning the city name, we have standardized the spelling of the city throughout the manuscript as “Chittagong”. To avoid confusion for international readers, we have added a clarifying statement noting that “Chittagong (also locally known as Chottogram) refers to the same city in Bangladesh.” When it first appeared.

Third, regarding educational qualifications, the reviewer suggested combining the “Higher secondary” and “University graduate” categories to be consistent with the questionnaire. We evaluated this suggestion but found that combining these groups would result in a disproportionate category, as nearly 70% of participants fall into these two groups. To preserve meaningful distinctions in educational attainment, we have retained the current categories in Tables 1 and 2 and throughout the manuscript.

Fourth, we acknowledge the reviewer’s concern regarding the six negatively worded items in the respectful maternity care (RMC) questionnaire, where answering “yes” indicates disrespectful care. We have clarified in the Methods section that these six items were reverse coded prior to dichotomization.

“For the six negatively worded items, responses were reverse coded prior to dichotomization so that “disagree” or “strongly disagree” indicated respectful care. This ensured that across all items, a response coded as “Yes” consistently represented respectful care, and “No” indicated disrespect or abuse. This categorization was applied for the overall scale as well as for each of the specific domains.” (Page 8, Line 181-188)

The results presented in the revised manuscript now accurately reflect these corrections. Additionally, the Methods section has been updated to clearly describe the coding procedure, ensuring that readers understand how RMC was classified prior to analysis.

We sincerely thank the reviewer again for these valuable comments, which have helped us ensure the accuracy, clarity, and scientific rigor of our manuscript.

Reviewer #3:

Thank you for submitting your manuscript. The following are some minor comments to help you improve the paper.

General & Title

Title: I suggest a revised title to better reflect the study's design and focus: "Factors associated with respectful maternity care during hospital deliveries: a cross-sectional study in Bangladesh."

Results Section

Clarity of Acronyms: For the first time they appear in both the abstract and the results section, please spell out aOR (Adjusted Odds Ratio) and 95% CI (95% Confidence Interval) before using the acronyms.

Figure 1: I recommend removing Figure 1 as it does not appear to be essential for the main findings of the paper.

Table 2: I suggest reorganizing your presentation of the data. Please remove the first two columns (Respectful maternity care) from Table 2. Instead, create a new table that presents all the predictor variables against the two outcome categories (respected vs. not respected), including p-values. This new table should be placed just before your logistic regression analysis.

Discussion Section

Limitations: Please include this point as one of key limitations of the study: the potential information loss that occurred when converting the original 5-point Likert scale into a binary (Yes/No) categorical variable.

Policy Implications: I recommend adding a section that discusses the practical policy implications of your findings, outlining how this research can inform future interventions or policies in Bangladesh.

Authors’ response:

We sincerely thank the reviewer for the thoughtful comments and suggestions, which have helped improve the clarity and impact of our manuscript.

We appreciate the reviewer’s suggestion regarding title. The manuscript title has been revised to “Factors associated with respectful maternity care during hospital deliveries: a cross-sectional study in Bangladesh” to better reflect the study design and focus.

In response to the comment on acronyms, we have now spelled out Adjusted Odds Ratio (aOR) and 95% Confidence Interval (95% CI) the first time they appear in both the abstract and the Results section, before using the abbreviated forms subsequently. Regarding Figure 1, we have removed it, as it was not essential to the main findings of the paper.

Following the reviewer’s recommendation, we have reorganized the presentation of the data. The first two columns (Respectful maternity care) have been removed, and a new table has been created that presents all predictor variables against the two outcome categories (respected vs. not respected), including p-values. This new table has been placed just before the logistic regression analysis to improve flow and clarity.

We have added a limitation regarding the potential information loss that could occur when converting the original five-point Likert scale into a binary (Yes/No) variable, noting that this may have reduced the granularity of participants’ responses.

Additionally, we have included a section on policy and practice implications, highlighting how the findings can inform future interventions and policies in Bangladesh. This section emphasizes the need for strengthening respectful maternity care in public hospitals, targeted training for healthcare providers, structured referral pathways, and embedding respectful care into national maternal health policies and quality assurance frameworks.

We sincerely thank the reviewer again for these valuable comments, which have enhanced the clarity, rigor, and practical relevance of our manuscript.

---

## [Decision Letter · Decision Letter 1]

16 Sep 2025

PONE-D-25-20339R1Factors associated with respectful maternity care during hospital deliveries: a cross-sectional study in BangladeshPLOS ONE

Dear Dr. Hasan,

Thank you for submitting your manuscript to PLOS ONE. After careful consideration, we feel that it has merit but does not fully meet PLOS ONE’s publication criteria as it currently stands. Therefore, we invite you to submit a revised version of the manuscript that addresses the points raised during the review process. Please submit your revised manuscript by Oct 31 2025 11:59PM. If you will need more time than this to complete your revisions, please reply to this message or contact the journal office at plosone@plos.org . Please include the following items when submitting your revised manuscript:

We look forward to receiving your revised manuscript.

Kind regards,

Paridhi Jha, PhD

Academic Editor

PLOS ONE

Journal Requirements:

Reviewers' comments:

Reviewer's Responses to Questions

**Comments to the Author**

1. If the authors have adequately addressed your comments raised in a previous round of review and you feel that this manuscript is now acceptable for publication, you may indicate that here to bypass the “Comments to the Author” section, enter your conflict of interest statement in the “Confidential to Editor” section, and submit your "Accept" recommendation.

Reviewer #2: All comments have been addressed

Reviewer #3: All comments have been addressed

2. Is the manuscript technically sound, and do the data support the conclusions?

Reviewer #2: Yes

Reviewer #3: Yes

3. Has the statistical analysis been performed appropriately and rigorously? 

Reviewer #2: Yes

Reviewer #3: Yes

4. Have the authors made all data underlying the findings in their manuscript fully available?

Reviewer #2: Yes

Reviewer #3: Yes

5. Is the manuscript presented in an intelligible fashion and written in standard English?

Reviewer #2: Yes

Reviewer #3: Yes

6. Review Comments to the Author

Reviewer #2: You answered all reviewers' questions well. I have a few small items, and then this is ready for publication, in my view.

Line 80 refers to Bangladesh as “a developing country of south-east Asian region.” Bangladesh is in South Asia, not Southeast Asia. The phrase “developing country” has been superseded by the specific income level of the country named. In the case of Bangladesh, it is a lower-middle-income country. Please change the phrase on line 80 to “a lower-middle-income country in South Asia.” Please review this World Bank list if needed: https://datahelpdesk.worldbank.org/knowledgebase/articles/906519-world-bank-country-and-lending-groups.

Thank you for the information about education levels. I’m noting that your study includes much higher percentages of women educated to upper secondary and university levels than Bangladesh generally. This likely tracks closely with socioeconomic status, and you mention in your limitations that you do not handle socioeconomic disparities in detail. That is reasonable (because most of the population you are capturing are those who can go to a private hospital) and so no change is needed there. However, you could also include a sentence or two in the discussion/limitation section noting that you captured a cohort that skews more highly educated than the female population of Bangladesh generally, and this limits how generalizable your findings are.

In Table 3, you use the term “cOR.” I assume you mean “crude odds ratio.” Please spell out the full term somewhere in the document.

Finally, Table 3 shows some dramatic shifts between your cOR and aOR values, including some changes in direction (eg, with respect to facilities referrals, from cOR of 0.56 (0.38, 0.83)* to aOR of 2.88 (1.59, 5.31), so changing from lower likelihood to higher likelihood of respectful care). Is this due to small sample sizes? Please review closely and ensure all values are correct, and please comment on these shifts in your discussion or limitation section.

Reviewer #3: Thank you for your revisions. I have reviewed the updated manuscript and find the changes to be satisfactory. I have no additional comments.

7. PLOS authors have the option to publish the peer review history of their article (what does this mean? ). If published, this will include your full peer review and any attached files.

**Do you want your identity to be public for this peer review?** For information about this choice, including consent withdrawal, please see our Privacy Policy .

Reviewer #2: No

Reviewer #3: No

---

## [Author Response · Author response to Decision Letter 2]

17 Sep 2025

Reviewer #2:

You answered all reviewers' questions well. I have a few small items, and then this is ready for publication, in my view.

Line 80 refers to Bangladesh as “a developing country of south-east Asian region.” Bangladesh is in South Asia, not Southeast Asia. The phrase “developing country” has been superseded by the specific income level of the country named. In the case of Bangladesh, it is a lower-middle-income country. Please change the phrase on line 80 to “a lower-middle-income country in South Asia.” Please review this World Bank list if needed: https://datahelpdesk.worldbank.org/knowledgebase/articles/906519-world-bank-country-and-lending-groups.

Thank you for the information about education levels. I’m noting that your study includes much higher percentages of women educated to upper secondary and university levels than Bangladesh generally. This likely tracks closely with socioeconomic status, and you mention in your limitations that you do not handle socioeconomic disparities in detail. That is reasonable (because most of the population you are capturing are those who can go to a private hospital) and so no change is needed there. However, you could also include a sentence or two in the discussion/limitation section noting that you captured a cohort that skews more highly educated than the female population of Bangladesh generally, and this limits how generalizable your findings are.

In Table 3, you use the term “cOR.” I assume you mean “crude odds ratio.” Please spell out the full term somewhere in the document.

Finally, Table 3 shows some dramatic shifts between your cOR and aOR values, including some changes in direction (eg, with respect to facilities referrals, from cOR of 0.56 (0.38, 0.83)* to aOR of 2.88 (1.59, 5.31), so changing from lower likelihood to higher likelihood of respectful care). Is this due to small sample sizes? Please review closely and ensure all values are correct, and please comment on these shifts in your discussion or limitation section.

Authors’ Response:

We sincerely thank the reviewer for the careful review of our manuscript and for the insightful and constructive comments.

First, we have updated line 80 to “a lower-middle-income country in South Asia” per your suggestion.

Second, we acknowledged that our cohort includes a higher proportion of women with upper secondary and university education than the general female population, which may limit generalizability; this has been added to the limitations section of the manuscript.

Third, the term “cOR” means “crude odds ratio (cOR)” which is now mentioned in the footnote of Table 3.

According to your suggestion, we reviewed the analysis and we agree that shifts between cOR and aOR, including reversals (e.g., facility referrals), likely reflect confounding and the relatively small sample size. This has been noted in the limitations section of the manuscript.

Finally, we greatly appreciate the reviewer’s thoughtful guidance in strengthening the manuscript.

Reviewer #3:

Thank you for your revisions. I have reviewed the updated manuscript and find the changes to be satisfactory. I have no additional comments.

Authors’ Response:

We sincerely thank the reviewer for taking the time to review our revised manuscript and for the positive feedback. We greatly appreciate your acknowledgment and support.

---

## [Decision Letter · Decision Letter 2]

28 Sep 2025

Factors associated with respectful maternity care during hospital deliveries: a cross-sectional study in Bangladesh

PONE-D-25-20339R2

Dear Dr. Hasan,

We’re pleased to inform you that your manuscript has been judged scientifically suitable for publication and will be formally accepted for publication once it meets all outstanding technical requirements.

Kind regards,

Tanya Doherty, PhD

Academic Editor

PLOS ONE

Additional Editor Comments (optional):

Reviewers' comments:

Reviewer's Responses to Questions

**Comments to the Author**

1. If the authors have adequately addressed your comments raised in a previous round of review and you feel that this manuscript is now acceptable for publication, you may indicate that here to bypass the “Comments to the Author” section, enter your conflict of interest statement in the “Confidential to Editor” section, and submit your "Accept" recommendation.

Reviewer #2: All comments have been addressed

Reviewer #3: All comments have been addressed

2. Is the manuscript technically sound, and do the data support the conclusions?

Reviewer #2: Yes

Reviewer #3: Yes

3. Has the statistical analysis been performed appropriately and rigorously? 

Reviewer #2: Yes

Reviewer #3: Yes

4. Have the authors made all data underlying the findings in their manuscript fully available?

Reviewer #2: Yes

Reviewer #3: Yes

5. Is the manuscript presented in an intelligible fashion and written in standard English?

Reviewer #2: Yes

Reviewer #3: Yes

6. Review Comments to the Author

Reviewer #2: No further recommendations. Thank you for your diligence, and congratulations on the publication of your research.

Reviewer #3: Factors associated with respectful maternity care during hospital deliveries: a cross-sectional study in Bangladesh. All comments have been addressed. I have no comments

7. PLOS authors have the option to publish the peer review history of their article (what does this mean? ). If published, this will include your full peer review and any attached files.

**Do you want your identity to be public for this peer review?** For information about this choice, including consent withdrawal, please see our Privacy Policy .

Reviewer #2: No

Reviewer #3: No

---

## [Editor Report · Acceptance letter]

PONE-D-25-20339R2

PLOS ONE

Dear Dr. Hasan,

I'm pleased to inform you that your manuscript has been deemed suitable for publication in PLOS ONE. Congratulations! Your manuscript is now being handed over to our production team.

Kind regards,

on behalf of

Professor Tanya Doherty

Academic Editor

PLOS ONE